# Observing a scale anomaly and a universal quantum phase transition in graphene

O. Ovdat[1], Jinhai Mao[2], Yuhang Jiang[2], E.Y. Andrei[2] & E. Akkermans[1]

One of the most interesting predictions resulting from quantum physics, is the violation of classical symmetries, collectively referred to as anomalies. A remarkable class of anomalies occurs when the continuous scale symmetry of a scale-free quantum system is broken into a discrete scale symmetry for a critical value of a control parameter. This is an example of a (zero temperature) quantum phase transition. Such an anomaly takes place for the quantum inverse square potential known to describe 'Efimov physics'. Broken continuous scale symmetry into discrete scale symmetry also appears for a charged and massless Dirac fermion in an attractive $1/r$ Coulomb potential. The purpose of this article is to demonstrate the universality of this quantum phase transition and to present convincing experimental evidence of its existence for a charged and massless fermion in an attractive Coulomb potential as realized in graphene.

[1] Department of Physics, Technion, Israel Institute of Technology, Haifa 3200003, Israel. [2] Department of Physics and Astronomy, Rutgers University, Piscataway, NJ 08854, USA. O. Ovdat and Jinhai Mao contributed equally to this work. Correspondence and requests for materials should be addressed to E.A. (email: eric@physics.technion.ac.il)

Continuous scale symmetry (CS)—a common property of physical systems—expresses the invariance of a physical quantity $f(x)$ (e.g., the mass) when changing a control parameter $x$ (e.g., the length). This property is expressed by a simple scaling relation, $f(ax) = bf(x)$, satisfied $\forall a > 0$ and corresponding $b(a)$, whose general solution is the power law $f(x) = Cx^\alpha$ with $\alpha = \ln b/\ln a$. Other physical systems possess the weaker discrete scale symmetry (DS) expressed by the same aforementioned scaling relation but now satisfied for fixed values $(a, b)$ and whose solution becomes $f(x) = x^\alpha G(\ln x/\ln a)$, where $G(u + 1) = G(u)$ is a periodic function. Physical systems having a DS are also known as self-similar fractals[1] (Fig. 1a). It is possible to break CS into DS at the quantum level, a result which constitutes the basis of a special kind of scale anomaly[2, 3].

A well-studied example is provided by the problem of a particle of mass $\mu$ in an attractive inverse square potential[4, 5], which plays a role in various systems[6–9] and more importantly in Efimov physics[10, 11]. Although well defined classically, the quantum mechanics of the scale—and conformal[12]—invariant Hamiltonian $H = -\Delta/2\mu - \xi/r^2$ (with $\hbar = 1$) is well posed, but for large enough values of $\xi$, $H$ is no longer self-adjoint[13, 14]. The corresponding Schrödinger equation for a normalisable wave function $\psi(r)$ of energy $k^2 = -2\mu E$ is,

$$\psi''(r) + \frac{d-1}{r}\psi'(r) + \frac{\zeta}{r^2}\psi(r) = k^2\psi(r), \tag{1}$$

where $\zeta \equiv 2\mu\xi - l(l + d - 2)$ is a dimensionless parameter, $d$ the space dimensionality and $l$ the orbital angular momentum. Equation (1) is invariant under the transformation $r \to \lambda r$ and $k \to k/\lambda$, $\forall\lambda$ (CS), namely to every normalisable wave function of energy $k^2$ corresponds a continuous family of states with energies $(\lambda k)^2$, so that the bound spectrum is a continuum unbounded from below. Various ways exist to cure this problem, based on cutoff regularisation and renormalisation group[15–21], and all lead for the low-energy spectrum to a quantum phase transition (QPT) monitored by $\zeta$, between a single bound state for $\zeta < \zeta_c$ to an infinite and discrete energy spectrum for $\zeta > \zeta_c$, independent

of the regularisation procedure and given by

$$k_n(\zeta) = \epsilon_0 e^{-\frac{\pi n}{\sqrt{\zeta - \zeta_c}}}, \; n \in \mathbb{Z}, \tag{2}$$

which clearly displays DS. The critical value $\zeta_c = (d - 2)^2/4$ depends on the space dimensionality only, and $\epsilon_0$ is a regularization dependent energy scale. In the overcritical phase $\zeta > \zeta_c$, the corresponding renormalization group solution provides a rare example of a limit cycle[15, 16, 22]. Building on the previous example, it can be anticipated that the problem of a massless Dirac fermion in an attractive Coulomb potential[23–25], $-Z\alpha/r$, is also scale invariant (CS) and that the spectrum of resonant quasi-bound states presents similar features and a corresponding QPT.

In this work, we demonstrate the existence of such a universal QPT for arbitrary space dimension $d \geq 2$ and independently of the short distance regularisation. We obtain an explicit formula for the low-energy fractal spectrum in the overcritical regime. In contrast to the Schrödinger case equation (1), the massless Dirac Hamiltonian displays an additional parity symmetry which may be broken by the regularisation. In that case, the degeneracy of the overcritical fractal spectrum is removed and two intertwined geometric ladders of quasi-bound states appear in the s-wave channel. All these features are experimentally demonstrated using a charged vacancy in graphene. We observe the overcritical spectrum and we obtain an experimental value for the universal geometric ladder factor in full agreement with the theoretical prediction. We also explain the observation of two intertwined ladders of quasi-bound states as resulting from the breaking of parity symmetry. Finally, we relate our findings to Efimov physics as measured in cold atomic gases.

## Results

**The Dirac model.** The Dirac equation of a massless fermion in the presence of a $-Z\alpha/r$ potential is obtained from the Hamiltonian (with $\hbar = c = 1$),

$$H = -i\gamma^0\gamma^j\partial_j - \frac{\beta}{r}, \tag{3}$$

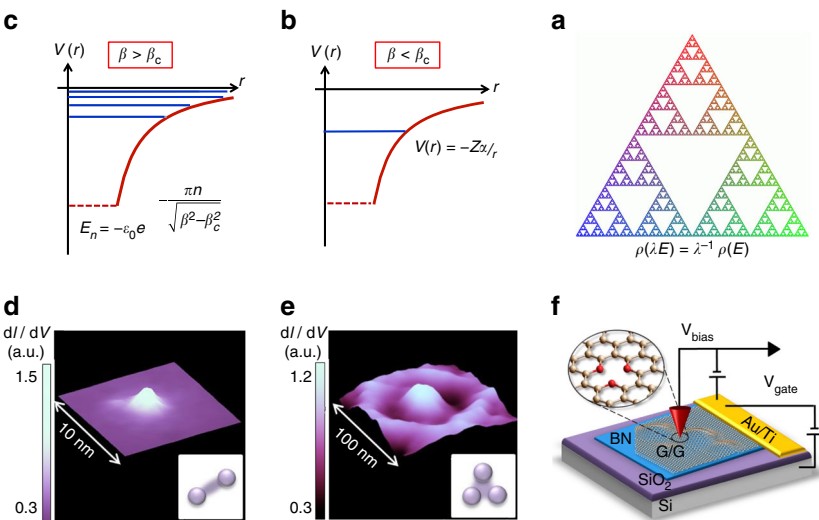

**Fig. 1** Schematic visualization of the purpose and main results of this paper. **a** Sierpinski gasket as typical featuring of such iterative fractal structures. This QPT is realized experimentally by creating single-atom vacancies in graphene. The function $\rho(E)$ is the density of states and obeys a scaling relation characterising the existence of discrete scale symmetry. **b, c** Illustration of the universal quantum phase transition (QPT) obtained by varying the dimensionless parameter $\beta \equiv Z\alpha$ (see text for precise definitions) in the low-energy spectrum of a massless fermion in a Coulomb potential $V = -Z\alpha/r$ created by a charge $Z$. **b** For low values, $\beta < \beta_c$, there is a single quasi-bound state close to zero energy. **c** For overcritical values, $\beta > \beta_c$, the low-energy spectrum is a ladder $E_n$ characterized by a discrete scale symmetry $\{E_n\} = \{\lambda E_n\}$ for $\lambda = \exp(\pi/\sqrt{\beta^2 - \beta_c^2})$. **d, e** Experimental d$I$/d$V$ maps of charged vacancy for fixed $\beta < \beta_c$ (**d**) and $\beta > \beta_c$ (**e**). The images illustrate the characteristic probability density of the resonances in (**b, c**). **f** Scanning tunnelling microscopy (STM) setup. Local charge $Z$ is accumulated at the single vacancy in graphene by applying voltage pulses to the STM tip

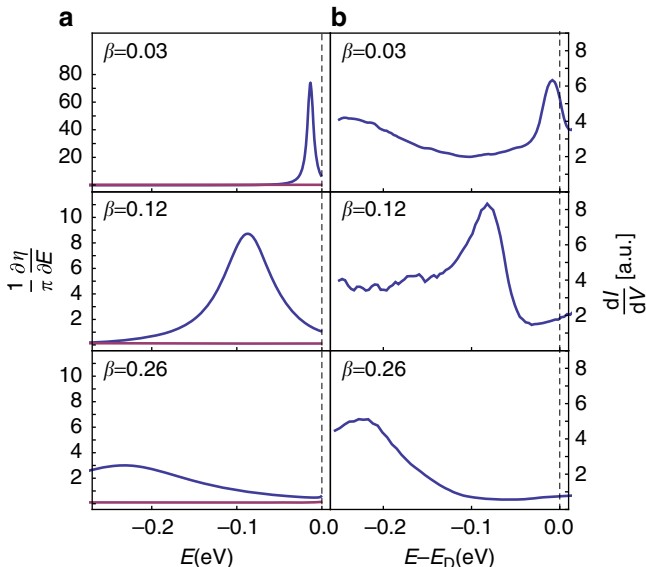

**Fig. 2** Experimental and theoretical picture in the undercritical regime. **a** Theoretical behavior of $(1)/(\pi)\mathrm{d}\eta/\mathrm{d}E$ for $d=2$ showing quasi-bound states of a massless Dirac fermion in the undercritical regime $\beta < 1/2$. In the scale-free low-energy, $EL \ll 1$ regime, the $m=-1$ (*blue*) branch contains a single peak and the $m=0$ (*purple*) branch shows no peak independently of the choice of boundary condition (see Supplementary Note 2). While increasing $\beta$, the resonance shifts to lower energy and becomes broader. **b** Excitation spectrum measured in graphene using STM as a function of the applied voltage $V$. The determination of the parameter $\beta$ is explained in the text

where $(\gamma^0, \gamma^j)$ are Dirac matrices. Here the dimensionless parameter monitoring the transition is $\beta = Z\alpha$, where $Z$ is the Coulomb charge and $\alpha$ the fine structure constant. The QPT occurs at the critical value $\beta_c = (d-1)/2$ (Supplementary Note 1) (A related anomalous behavior in the Dirac Coulomb problem has been identified long ago[26] but its physical relevance was marginal since it required non existent heavy-nuclei Coulomb charges $Z \simeq 1/\alpha \simeq 137$ to be observed. Moreover, the problem of a massive Dirac particle is different due to the existence of a finite gap which breaks CS.). For resonant quasi-bound states, we look for scattering solutions of the form $\psi_{in} + e^{2i\eta}\psi_{sc}$, where $\eta(E)$ is the energy-dependent scattering phase shift and $\psi_{in,sc}(r, E)$ are two component objects representing the radial part of the Dirac spinor which behave asymptotically as,

$$\psi_{in,sc}(r, E) = r^{\frac{1-d}{2}}\left(V_{in,sc}\,(2i|E|r)^{\mp i\beta}\,e^{\mp iEr}\right) \quad (4)$$

for $|E|r \gg 1$ and, using $\gamma \equiv \sqrt{\beta^2 - \beta_c^2}$,

$$\psi_{in,sc}(r, E) = r^{\frac{1-d}{2}}\left(U_{in,sc}^-\,(2iEr)^{-i\gamma} + U_{in,sc}^+\,(2iEr)^{i\gamma}\right), \quad (5)$$

for $|E|r \ll 1$ and for the lowest angular momentum channels. The two component objects $V_{in,sc}$ and $U_{in,sc}^\pm$ in Eqs. (4) and (5) are constants. It is easy to infer from (5) that $\beta = \beta_c$ plays a special role. Indeed for $\beta > \beta_c$, there exists a family of normalisable solutions that admit complex eigenvalues $E = -i\epsilon$, hence the Hamiltonian (3) is not self-adjoint ($H \neq H^\dagger$). To properly define this quantum problem, a regularisation is thus needed for the too strong potential at overcritical values of $\beta = Z\alpha$. This is achieved by introducing a cutoff length $L$ and a boundary condition at $r=L$, which is equivalent to replacing the Coulomb potential at short distances by a well-behaved potential whose exact form is irrelevant in the low-energy regime $EL \ll 1$.

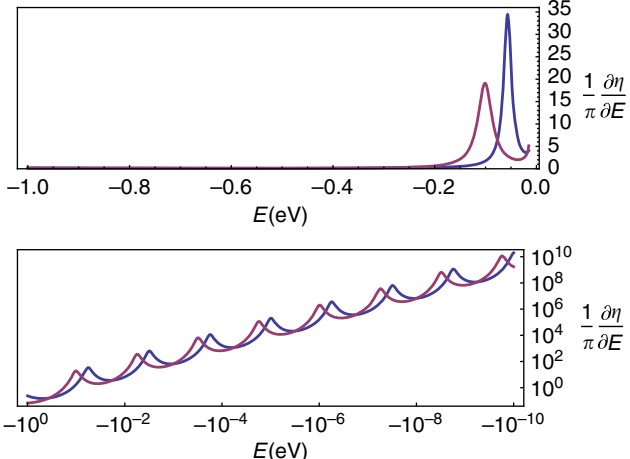

**Fig. 3** Theoretical behavior of the low energy and scale-free part of the quasi-bound states spectrum in the overcritical regime for $d=2$ and $\beta = 1.2 > \beta_c(= 1/2)$. The lower plot displays the detailed structure of the infinite geometric ladders. Note that the $m=-1$ (*blue*) and $m=0$ (*purple*) ladders are intertwined. These results are independent of the boundary condition

The resulting mixed boundary condition can be written as $h = \Psi_2(r, E)/\Psi_1(r, E)|_{r\to L^+}$, where $\Psi_{1,2}$ represent the two components of the aforementioned radial part of the Dirac spinor. The resulting scattering phase shift $\eta(E, L, h)$, which contains all the information about the regularisation, thus becomes a function of $L$ and of the parameter $h$. The quasi-bound states energy spectrum is obtained from the scattering phase shift by means of the Krein–Schwinger relation[27, 28] which relates the change of density of states $\delta\rho$ to the energy derivative of $\eta$, (This is also related to the Wigner time delay[29] and to the Friedel sum rule)

$$\delta\rho(E) = \frac{1}{\pi}\frac{\mathrm{d}\eta(E)}{\mathrm{d}E}. \quad (6)$$

**Theoretical structure of quasi-bound spectrum.** From now on, and to compare to experimental results further discussed, we consider the case $d=2$, for which there is a single orbital angular momentum quantum number $m \in \mathbb{Z}$. The corresponding critical coupling becomes $\beta_c = |m + 1/2| \geq 1/2$, giving rise to the $s$-wave channels, $m=0, -1$, for which $\beta_c = 1/2$. Depending on the choice of boundary condition $h$, $\delta\rho(E)$ can be degenerate or non-degenerate over these two $s$-wave channels. This degeneracy originates from the symmetry of the $(2+1)$ Dirac Hamiltonian (3) under parity, $(x, y) \to (-x, y)$, and its existence is equivalent to whether or not the boundary condition breaks parity (Supplementary Note 2). In what follows, we will consider the generic case in which there is no degeneracy. In the undercritical, $\beta < \beta_c$, and low-energy regime $EL \ll 1$, we observe (Figs. 1b, 2a) a single quasi-bound state originating from only one of the $s$-wave channels and which broadens as $\beta$ increases. In the overcritical regime $\beta > \beta_c$, this picture changes dramatically. (We emphasize that this picture remains valid for all values of $\beta > \beta_c$ and not only in the vicinity of $\beta_c$.) The low-energy ($EL \ll 1$) scattering phase shift displays two intertwined, infinite geometric ladders of quasi-bound states (Figs. 1c, 3) at energies $E_n$ still given by (2) but with $\zeta - \zeta_c$ now replaced by $\beta^2 - \beta_c^2$. (Moreover, note that the energy scale $\epsilon_0$ for the Dirac case is different from the inverse square Schrödinger case defined in equation (1)). This sharp transition at $\beta_c$ belongs to the same universality class as presented for the inverse square Schrödinger problem, namely CS

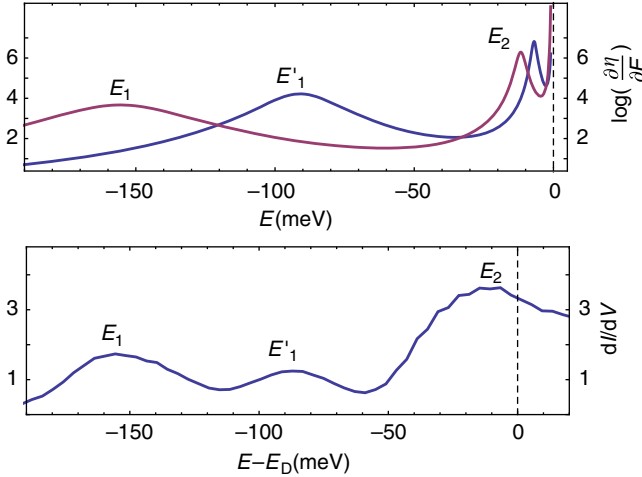

**Fig. 4** Experimental and theoretical picture in the overcritical regime. *Upper plot*: Theoretical behavior of the low energy and scale-free part of the overcritical ($\beta = 1.33$) quasi-bound states spectrum obtained from (6). The blue (*purple*) line corresponds to $m = -1$ ($m = 0$). *Lower plot*: Experimental values of the (STM) tunnelling conductance measured at the position of charged vacancies in graphene. The labelling $E_1$, $E_2$, $E'_1$ of the peaks is explained in the text

of the quasi-bound states spectrum is broken for $\beta > \beta_c$ into a DS phase characterized by a fractal distribution of quasi-bound states. The QPT thus reflects the lack of self-adjointness of the Hamiltonian equation (3) and the necessary regularisation procedure leads to a scale anomaly in which CS is broken into DS.

**Experimental realization in graphene.** A particularly interesting condensed matter system, where the previous considerations seem to be relevant is graphene in the presence of implanted Coulomb charges in conveniently created vacancies[30]. It is indeed known that low-energy excitations in graphene behave as a massless Dirac fermion field with a linear dispersion $\epsilon = \pm v_F |p|$ and a Fermi velocity $v_F \simeq 10^6$ m/s[31]. These characteristics have been extensively exploited to make graphene a very useful platform to emulate specific features of quantum field theory, topology and especially QED[23], since an effective fine structure constant $\alpha_G \equiv e^2/\hbar v_F$ of order unity is obtained by replacing the velocity of light $c$ by $v_F$.

It has been recently shown that single-atom vacancies in graphene can stably host local charge[30]. Density functional theory calculations have shown that, when a carbon atom is removed from the honeycomb lattice, the atoms around the vacancy site rearrange into a lower energy configuration[32]. The resulting lattice reconstruction causes a charge redistribution, which in the ground state has an effective local charge of $\approx +1$. Recent Kelvin probe force microscopy measurements of the local charge at the vacancy sites are in good agreement with the Density functional theory predictions. Vacancies are generated by sputtering graphene with $He^+$ ions[33, 34]. Charge is modified and measured at the vacancy site by means of scanning tunnelling spectroscopy and Landau level spectroscopy as detailed in ref. [30]. Applying multiple pulses allows for a gradual increase in the vacancy charge, which in turn acts as an effective tunable Coulomb source. Moreover, the size of the source inside the vacancy is small ($\approx 1$ nm) as compared to the method of deposited metal clusters[35]. Using this method, we are able to observe the transition expected to occur at $\beta = 1/2$ and to measure and analyze three resonances for a broad range of $\beta$ values.

To establish a relation between the measured differential conductance and the spectrum of quasi-bound states, we recall

that the tunnel current $I(V)$ is proportional to both the density of states $\rho_t(\epsilon)$ of the STM tip and $\rho(\epsilon)$ of massless electronic excitations in graphene at the vacancy location. We also assume that the tunnel matrix element $|t|^2$ depends only weakly on energy and that both voltage and temperature are small compared to the Fermi energy and height of the tunnelling potential, so that the current $I(V) = G_t V$ is linear with $V$ thus defining the tunnel conductance $G_t = 2\pi(e^2/\hbar)|t|^2 \rho_t(\epsilon)$. Assuming that $\rho_t$ of the reference electrode (the tip) is energy independent, a variation $\delta\rho(\epsilon)$ of the local density of states at the vacancy leads to a variation $\delta I(V)$ of the current and thus to a variation $\delta G_t(V)$ of the tunnel conductance so that, at zero temperature, we obtain[36]

$$\frac{\delta G_t(V)}{G_t} = \frac{\delta\rho(\epsilon)}{\rho_0}, \qquad (7)$$

where $\rho_0$ is the density of states in the absence of vacancy. By considering the vacancy as a local perturbation, each quasi-particle state is characterized by its scattering phase shift taken to be the phase shift $\eta(E)$ of the quasi-bound Dirac states previously calculated. Then, the change of density of states $\delta\rho(E)$ is obtained from equation (6) and combining together with equation (7) leads to the relation,

$$\frac{d\delta I}{dV} = \frac{G_t}{\pi\rho_0}\frac{d\eta(E)}{dE} \qquad (8)$$

between the differential tunnel conductance and the scattering phase shift.

The measurements and the data analysis presented here were carried out as follows: positive charges are gradually injected into an initially prepared single atom vacancy and the differential conductance $\delta G_t(V)$ is measured at each step as a function of voltage. Since we are looking at the positions of resonant quasi-bound states, both quantities displayed in Figs. 2, 4 give the same set of resonant energies, independently of the energy-independent factor $G_t/\pi\rho_0$. For low enough values of the charge, the differential conductance displayed in Fig. 2b, shows the existence of a single quasi-bound state resonance. The behavior close to the Dirac point, namely in the low-energy regime independent on the short distance regularization, is very similar to the theoretical prediction of Fig. 2a. When the build up charge exceeds a certain value, we note the appearance of three resonances, emerging out of the Dirac point. We interpret these resonances as the lowest overcritical ($\beta > 1/2$) resonances, which we denote $E_1$, $E'_1$, $E_2$, respectively. The corresponding theoretical and experimental behaviors displayed in Figs. 3, 4, show a very good qualitative agreement. To achieve a quantitative comparison solely based on the previous Dirac Hamiltonian equation (3), we fix $L$ and the boundary condition $h$ and deduce the theoretical $\beta$ values corresponding to the respective positions of the lowest overcritical resonance $E_1$ (as demonstrated in Fig. 4). This allows to determine the lowest branch $E_1(\beta)$ for $n = 1$ represented in Fig. 5. Then, the experimental points $E'_1$, $E_2$ are directly compared to their corresponding theoretical branch as seen in Fig. 5. We determine $L$ and $h$, according to the ansatz $h = a(m + 1)$, and obtain the best correspondence for $L \simeq 0.2$ nm, $a \simeq -0.85$. We compare the experimental $E_2/E_1$ ratio with the universal prediction $E_{n+1}/E_n = e^{-\pi/\sqrt{\beta^2 - 1/4}}$ as seen in Fig. 6. A trend-line of the form $e^{-b/\sqrt{\beta^2 - 1/4}}$ is fitted to the ratios $E_2/E_1$, yielding a statistical value of $b = 3.145$ with standard error of $\Delta b = 0.06$ consistent with the predicted value $\pi$. An error of $\pm 1$ mV is assumed for the position of the energy resonances.

A few comments are appropriate: (i) The points on the $E_2(\beta)$ curve follow very closely the theoretical prediction $E_{n+1}/E_n = e^{-\pi/\sqrt{\beta^2 - 1/4}}$. This result is insensitive to the choice

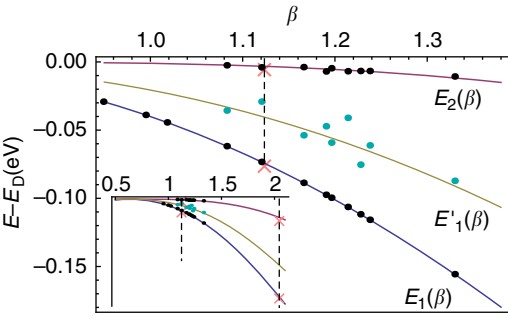

**Fig. 5** Behavior of the energies $E_n(\beta)$ of the quasi-bound state spectrum. The curves are obtained from (2) for $E_1(\beta)$, $E'_1(\beta)$, $E_2(\beta)$ as adapted to the massless Dirac case. The *black* and *cyan dots* correspond to the values measured in graphene. The two *pink x*'s are the values of Efimov energies measured in Caesium atoms[39, 47], which corresponds to the (overcritical) fixed Efimov value $\beta_E = 1.1236$. Additional experimental points obtained in refs [40, 41] are displayed in the *inset*

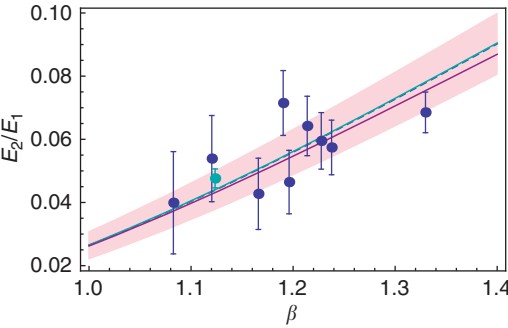

**Fig. 6** Comparison between the experimentally obtained $E_2/E_1$ ratio and the universal factor $e^{-\pi/\sqrt{\beta^2-1/4}}$. *Blue points*: the ratio $E_2/E_1$ obtained from the position of the points in Fig. 5. Cyan point: Universal Efimov energy ratio as measured in Caesium atoms[39, 47]. *Blue line* (dashed): the corresponding optimized curve, fitted according to the model $e^{-b/\sqrt{\beta^2-1/4}}$ and corresponding to $b = 3.145$ with standard error of $\Delta b = 0.06$ consistent with the predicted value $\pi$. The *shaded pink* region is the $\pm 2\Delta b$ confidence interval of the curve. *Cyan* line: universal low-energy factor $e^{-\pi/\sqrt{\beta^2-1/4}}$. *Purple line*: theoretical ratio $E_2/E_1$ obtained from the exact solution of the Dirac equation. As $\beta \to 0.5$, $|E_n|$ becomes smaller therefore the green and purple curves coincide for low $\beta$. The *error bar* on the resonance energies is $\pm 1$ mV

of $h$, thus manifesting the universality of the ratio $E_{n+1}/E_n$. (ii) In contrast, the correspondence between the $E'_1$ points and the theoretical branch is sensitive to the choice of $h$. This reflects the fact that while each geometric ladder is of the form equation (2) (with the appropriate $\zeta \to \beta$ change), the energy scale $\epsilon_0$ is different between the two thus leading to a shifted relative position of the two geometric ladders in Fig. 3. The ansatz taken for $h$ is phenomenological (Supplementary Note 2), however, we find that in order to get reasonable correspondence to theory, the explicit dependence on $m$ is needed. More importantly, it is necessary to use a degeneracy breaking boundary condition to describe the $E'_1(\beta)$ points. For instance, if the Coulomb potential is regularized by a constant potential for $r \leq L$[37], then both angular momentum channels (i.e., the $E'_1$ and $E_1$ points) become degenerate. The existence of the experimental $E'_1$ branch is therefore a distinct signal that parity symmetry in the corresponding Dirac description equation (3) is broken. In graphene, exchanging the triangular sublattices is equivalent to a parity transformation. Creating a vacancy breaks the symmetry between the two sub-lattices and is therefore at the origin of broken parity in the Dirac model. (iii) The value $L \simeq 0.2$ nm is

fully consistent with the low-energy requirement $E_1 L/\hbar v_F \simeq 0.03 \ll 1$ necessary to be in the regime relevant to observe the $\beta$-driven QPT.

## Discussion

A further argument in support of the universality of this QPT is achieved by comparing the experimental results obtained in graphene with those deduced from a completely different physical problem. To that purpose, we dwell for a short while recalling the basics underlying Efimov physics[38]. Back to 1970, Efimov[10] studied the quantum problem of three identical nucleons of mass $m$ interacting through a short range ($r_0$) potential. He pointed out that when the scattering length $a$ of the two-body interaction becomes very large, $a \gg r_0$, there exists a scale-free regime for the low-energy spectrum, $\hbar^2/ma^2 \ll E \ll \hbar^2/mr_0^2$, where the corresponding bound-states energies follow the geometric series $\left(\sqrt{-E_n} = -\tilde{\epsilon}_0 e^{-\pi n/s_0}\right)$, where $s_0 \simeq 1.00624$ is a dimensionless number and $\tilde{\epsilon}_0$ a problem-dependent energy scale. Efimov deduced these results from an effective Schrödinger equation in $d = 3$ with the radial ($l = 0$) attractive potential $V(r) = -(s_0^2 + 1/4)/r^2$. Using Eqs. (1) and (2) and the critical value $\zeta_c = (d - 2)^2/4 = 1/4$ for this Schrödinger problem, we deduce the $\zeta$ value for the Efimov effect to be $s_0^2 + 1/4 > \zeta_c$ corresponding to the overcritical regime of the QPT. The value of $\beta$ matching to the Efimov geometric series factor $e^{\pi/s_0}$ is $\beta_E = \sqrt{s_0^2 + 1/4} = 1.1236$, referred to as the fixed Efimov value. Despite being initially controversial, Efimov physics has turned into an active field especially in atomic and molecular physics where the universal spectrum has been studied experimentally[39–46] and theoretically[38]. The first two Efimov states $E_n$ ($n = 1$, 2) have been recently determined using an ultracold gas of caesium atoms[47]. Although the Efimov spectrum always lies at a fixed and overcritical value of the coupling, unlike the case of graphene where $\beta$ can be tuned, the universal character of the overcritical regime allows nevertheless for a direct comparison of these two extremely remote physical systems. To that purpose, we include the Efimov value $\beta_E$ in the expression obtained for the massless Dirac fermion in a Coulomb potential and insert the corresponding data points obtained for cold atomic caesium in the graphene plot (Fig. 5) up to an appropriate scaling of $\tilde{\epsilon}_0$. The results are fully consistent thus showing in another way the universality presented.

There are other remote examples of systems displaying this universal QPT, e.g., flavoured QED3[48], and the XY model (Kosterlitz-Thouless[8] and roughening transitions[22]). Our results provide a useful and original probe of characteristic features of this universal QPT and motivate a more thorough study of this transition.

## Methods

Our sample is stacked two layers of graphene on top of a thin BN flake (see Fig. 1f). The standard dry transfer procedure is followed to get this heterostructure. A large twisted angle between the two layers graphene is selected in order to weaken the coupling. The free-standing like feature for the top layer graphene is checked by the Landau levels spectroscopy. To achieve the diluted single vacancies, the sample is exposed to the helium ion beam for short time (100 eV for 5 s) followed by the high temperature annealing. The experiment is performed at 4.2 K with a home-built STM. The $dI/dV$ ($I$ is the current, $V$ is the bias) is recorded by the standard lock-in technique, with a small AC modulation 2 mV at 473.1 Hz added on the DC bias. To tune the effective charge on the vacancy, we apply the voltage pulse (−2 V, 100 ms) with the STM tip directly locating on top of the vacancy.

**Data availability**. The data that support the findings of this study are available from the corresponding author upon request.

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

## Acknowledgements

This work was supported by the Israel Science Foundation Grant No. 924/09. Funding for the experimental work provided by DOE-FG02-99ER45742 (STM/STS), NSF DMR 1207108 (fabrication and characterization).

## Author contributions

O.O. and E.A.: Proposed observing the aforementioned quantum phase transition in graphene with a charged vacancy. They have contributed to interpreting and solving the theoretical model as well as analysing the experimental data and making contact with the theory. J.M., Y.J. and E.Y.A.: Conceived of and designed the experiment, as well as performed the measurements and analyzed the data. All authors contributed to discussions and preparation of the manuscript.
