## [Peer Review File · Nature Communications]

Reviewers' comments:

Reviewer #1 (Remarks to the Author):

"Collapse of the wave function" whereby solutions of the Schrodinger problem in a sufficiently strong Coulomb field require a short-distance regularisation in order to be defined, leading to a distinctive energy spectrum which is essentially singular in the coupling strength as the critical value is approached, has been a subject of theoretical interest for many decades. Because a renormalisation condition is required to yield physically sensible solutions, it is argued that requiring the insensitivity of the spectrum to the regularisation details yields universal predictions. Graphene provides a natural arena to expose the phenomenon in experiment, since its low-energy excitations obey a massless Dirac equation, and the underlying crystal lattice provides a natural cutoff.

This paper reviews the properties of the Dirac equation solutions in two spatial dimensions in both sub- and super-critical regimes, and then presents results obtained using STM in the vicinity of a charged vacancy, where crucially the vacancy charge can be varied; variations in the tunnelling current are mapped to the bound-state spectrum, and the variation of the ratio of the observed levels with coupling strength is shown to follow the theoretical predictions nicely. In support of universality it is also argued that similar solutions have been observed in ultra cold atomic systems, but in this case the variation with coupling is inaccessible.

In my view this is an important piece of work which stands comparison with experimental demonstrations of Klein tunnelling and zitterbewegung in graphene. I believe it is ultimately suitable for publication in Nature Communications. Before final acceptance, the authors should consider the following minor comments:

- (i) in the legend of Fig. 1(b) there is a minus sign missing from the argument of the exponential
- (ii) in eqns. (4) and (5) it should be mentioned that U_{\pm} and V are constant 2-spinors
- (iii) in measuring the spectrum using eqn. (8) to yield Figs. 2 & 4, are the data taken at the location of the maxima shown in Figs. 1(d),(e)? It would help if this were clarified.
- (iv) the solution (2) differs from the solution SM(15) (used to generate the curves in Fig. 5) not just by the substitution of beta squared for zeta, but also by a factor of 2 in the numerator of the exponent. Please check whether this is correct, and clarify this point in the text.
- (v) the fixed Efimov value is $\zeta_E = (1.1236)^2$
- (vi) below SM(15) in the Supplementary material it is stated that in the subcritical regime "there are no quasi bound states arbitrarily close to zero energy", in apparent contradiction with Fig. 1(a).

My only substantive point is that since the $E_1(\beta)$ points fit the curve by construction, a more honest, if less compelling, graph than Fig. 5 would plot energy ratios rather than having an absolute energy scale along the y-axis -- indeed this is the underlying point behind universality. Also, I am unable to follow how the Efimov data has been transferred to the graph unless the "problem-dependent" energy scale ϵ has been factored out. The authors should be clearer about their procedure. Also, why are the extra data from the Li-Cs system not included on the plot except in the Supplementary Material? - surely this would only increase the impact?

Reviewer #2 (Remarks to the Author):

The manuscript by Ovdad et al is well written and interesting to read. I do think it is a high quality work overall. However I must comment on the interpretations with "QPT" etc. The measured results did show remarkable changes with the tailoring of local potentials, but this may not correspond to the speculated QPT. Firstly, with the bilayer graphene particularly the locally damaged graphene, Dirac physics is no longer valid. Secondly, the authors should include an

actual atomic resolution STM scan to show how the atom positions actually look like in the sub critical and over critical situations, then the readers can easily judge whether the deformation is indeed a local confinement or some other wired form of classical confinement (which certainly will show similar change from a shallow trap with one state to deeper trap with multiple states). Thirdly, I am not sure how the local charge is created. Since graphene is conducting, the local charge should disappear as soon as the tip voltage is removed, unless the charge is actually injected into the SiO₂ substrate then it can be long lived and generate the readily tuneable Coulomb potentials.

Reviewer #3 (Remarks to the Author):

Review report for NCOMMS-17-01883-T

The manuscript reports on evidence for the observation of a quantum phase transition (QPT) that occurs when a continuous scale invariance is reduced to a discrete one. The evidence provided are STM spectroscopy measurements on graphene with a vacancy that introduces a local Coulomb potential.

The interpretation in terms of a QPT is novel and appears to qualitatively agree with the data presented. However, a careful reading leads to several questions that need to be addressed by the authors.

Regarding the experimental part: As mentioned above, the experimental setup is a charged vacancy in a graphene sheet that realizes an attractive Coulomb potential for electrons. The procedure to reach the conditions of the phase transition requires further charging of the impurity to increase the effective strength of the potential. The increase is reached by applying multiple current pulses with the STM tip. A relevant question is: what occurs with screening? (incidentally: there is no reference accompanying the statement that 'single atom vacancies in graphene can stably host local charge').

Regarding the theory interpretation there are two important points that need to be discussed: 1) the model is based on the analysis of a single (non-interacting) relativistic particle in terms of the Dirac equation in the presence of a Coulomb potential. The Coulomb potential that appears in Eq. 3 in the main text is proportional to $1/r$, where r is the radial coordinate in a polar coordinate system used to describe a 2d membrane. An electrical charge in graphene (either produced by a localized vacancy or an atom deposited on top or below) produces a 3d Coulomb potential (since it lives in a 3d world). I see two ways in which to consider this point: a) consider 'standard' spherical coordinates for the Coulomb potential and by proper integration obtain the effective 2d potential experienced by electrons in graphene, or b) use the generalized polar decomposition for cartesian coordinates and separate radial from angular dependence, fixing the appropriate angular momenta eigenvalues by the constraint of 2d. Either of these procedures would render a different dependence on the polar radial/angular coordinates than the one assumed in the model. Furthermore, although graphene is prepared on top of a substrate with a large twisted angle to achieve the weak coupling regime, there is screening produced by the substrate that is not even mentioned (I assume vacuum on top, thus the total screening is anisotropic).

2) the role played by electron-electron interactions that are completely neglected. There seems to be a disconnect between the mathematical model that deals with a finite number of single particle bound states (akin to the spectra for trions or other similar charged impurities in semiconductors) and the experimental setup that creates a charged region of reduced dimensions (akin to a quantum dot or island). It would be expected that electron-electron interactions play an important role in experiment but there are not addressed by the theory model. Is screening the reason why they can be neglected? if not, why their effect is not included in the model?

These considerations suggest that the form proposed for the interaction term is too simplistic to represent the experimental situation.

In addition to these points, there are other minor ones that deserve further attention:

- a) It is not clear why the Sierpinski gasket is included in Fig. 1. I assume it is for pedagogical purposes but it would be more useful if instead more details in the experimental data are included. For example: Is the amount of charge transferred to the impurity controlled by the duration of the pulse? or is it the intensity of the current? How is the amount of charge transferred known?
- b) Being such a local defect, the vacancy must break the valley degeneracy locally, why it is assumed that parity remains a good symmetry for the interpretation of the data? It is natural to wonder if the degeneracy condition that needs to be broken for the peaks at E'_1 is not due to the breaking of valley degeneracy.
- c) It is not clear what the STM images appearing in Fig. 1 represent. Do they represent the combined orbitals proposed in the theory model? If shown from the top would they look like the insets? The two figures have very different length scales: do they show the existence of a small 'charged island' after the transition is achieved? What is the reason for the Mexican hat-like structure?
- d) The comparison with the Efimov states with data from cold atom experiments is unclear. In the case of Efimov states the results are obtained for a 3d potential, while the fitting is done with a 2d potential. What is the nature of the connection?

Finally, the supplementary information presents a detailed account of the model and its solution and it is a welcome addition to the manuscript.

Overall the interpretation presented for the data is novel and interesting, however not persuasive enough to warrant publication in Nature Communications.

Reviewers' comments:

Reviewer #1 :

1. **in the legend of Fig. 1(b) there is a minus sign missing from the argument of the exponential**

We have implemented this change.

2. **in eqns. (4) and (5) it should be mentioned that U_{\pm} and V are constant 2-spinors**

We have added a sentence: "The two-component objects $V_{\text{in,sc}}$ and $U_{\text{in,sc}}^{\pm}$ in equations (4), (5) are constant." (page 2, right column) Note that we prefer not to use here the word 'spinor' since equations (4) and (5) are valid for a general space dimension d and for $d \neq 2$ these equations do not involve Dirac spinors but only represent the radial part of the full $d + 1$ dimensional Dirac spinor.

3. **in measuring the spectrum using eqn. (8) to yield Figs. 2 & 4, are the data taken at the location of the maxima shown in Figs. 1(d),(e)? It would help if this were clarified.**

Yes. The STM measurements are taken at the vacancy site. To make this clearer, we changed: 'Charge is created at the vacancy site and monitored by means of a scanning tunnelling microscope (STM) also used to measure the local change of (tunnelling) electronic density of states expressed by the behaviour of the differential conductance as a function of the applied voltage V ' to: 'Charge is modified and measured at the vacancy site by means of scanning tunnelling spectroscopy and Landau level spectroscopy as detailed in reference [30]' (page 4, left column)

4. **the solution (2) differs from the solution SM(15) (used to generate the curves in Fig. 5) not just by the substitution of beta squared for zeta, but also by a factor of 2 in the numerator of the exponent. Please check whether this is correct, and clarify this point in the text.**

We agree. There is a factor of 2 coming from the two different dispersion relations $E \sim p^2$ for Schrödinger and $E \sim p$ for Dirac. To make things consistent, we have changed in the text, the energy E_n in equation (2) to $k_n \equiv \sqrt{-2\mu E_n}$ where k is properly defined above equation (1).

5. **the fixed Efimov value is $\zeta_E = (1.1236)^2$**

As seen in figure 5, the fixed Efimov value is the number obtained while replacing β within $E_n/E_{n+1} = e^{\frac{\pi}{\sqrt{\beta^2 - \beta_c^2}}}$ so as to obtain the Efimov geometric factor $e^{\frac{\pi}{1.00624}} \approx 22.7$. Since $\beta_c = 1/2$, the solution is $\beta_E = \sqrt{s_0^2 + 1/4} = 1.1236$.

However, we agree that this is not clear from the text and we rephrased this part accordingly (page 5, right column). Specifically, we replaced the notation ζ_E by β_E .

6. **below SM(15) in the Supplementary material it is stated that in the sub-critical regime "there are no quasi bound states arbitrarily close to zero energy", in apparent contradiction with Fig. 1(a).**

Indeed, when both β and L are **fixed**, there are no quasi bound states arbitrarily close to zero energy as opposed to the case of a fixed $\beta > 1/2$ where there are infinitely many quasi-bound states arbitrarily close to zero energy. However, by taking the limit $L \rightarrow \infty$, the position of the under critical quasi-bound state goes to 0. Fig. 1(a) has been plotted in this limit and this is why the quasi-bound state shows up on the zero energy line.

We agree that this is confusing and we have changed the plot so that now it corresponds to a finite value of L where the quasi bound state is at a finite negative energy. In addition, we rephrased "there are no quasi-bound states arbitrarily close to zero energy" to "for fixed L and β , there are no quasi-bound states arbitrarily close to zero energy" (page 2 of SM, left column).

7. **since the $E_1(\beta)$ points fit the curve by construction, a more honest, if less compelling, graph than Fig. 5 would plot energy ratios rather than having an absolute energy scale along the y-axis – indeed this is the underlying point behind universality.**

We have taken this remark in serious consideration. To that purpose, we have re-analysed the experimental data and present them on a new additional graph (Fig. 6) where we quantitatively compare the energy ratios to the universal factor. In addition we added to the main text a new paragraph corresponding to this plot (page 4, right column).

While preparing this new graph, we have calculated approximate error bars of $2mV$ for each point in Fig. 5 and redetermined the position of the Dirac points and of the quasi bound state peaks within these intervals. Minor changes thus show up in Fig. 5 as a result of these considerations. This figure is important to describe the intermediate E'_1 branch and to have a general qualitative picture of our results.

8. **Also, I am unable to follow how the Efimov data has been transferred to the graph unless the "problem-dependent" energy scale epsilon tilde has been factored out. The authors should be clearer about their procedure.**

Indeed, an overall and problem-dependent energy scale is factored and it has been scaled out so as to match the problem-dependent energy scale of the Dirac case. What is universal is the geometric series structure of the spectrum in both problems presenting a discrete scale invariant behaviour in the low energy regime. Our purpose in inserting the cold atoms Efimov data points, is to emphasize both the universality and the fixed value of β in Efimov physics in a transparent and visual way. To make this clearer, we rephrased

“...insert the corresponding data points obtained for cold atomic caesium in the graphene plot (Fig. 5) without any further calibration.“

to

“insert the corresponding data points obtained for cold atomic caesium in the graphene plot (Fig. 5) (up to an appropriate scaling of $\tilde{\epsilon}_0$).” (page 5, right column)

9. **Also, why are the extra data from the Li-Cs system not included on the plot except in the Supplementary Material?- surely this would only increase the impact?**

We have indeed added these extra data in the Li-Cs system in an inset in Fig. 5 of the main text. Initially, we did not include these points because they are placed at a large value $\beta = 2.04$. Including them is at the expense of the visibility of the measured graphene data. The adding of the new Fig. 6 makes this issue less important.

Reviewer #2 :

1. **Firstly, with the bilayer graphene particularly the locally damaged graphene, Dirac physics is no longer valid.**

The first part of the reviewer’s question pertains to the validity of Dirac physics in double layer graphene. If this was bilayer graphene, as opposed to a double layer consisting of two superposed single layers, the dispersion would be quadratic in which case, as the reviewer correctly points out, the Dirac physics would have been lost. However, as was shown both experimentally [1–3] and theoretically [4], the spectrum of double layer graphene is qualitatively different from that of the bilayer. Importantly, its structure is a function of the twist angle, θ , between the crystallographic orientations of the two layers. For sufficiently large angles, $\theta > 10^\circ$, which is the case of the double layer reported here, the low energy spectrum is indistinguishable from that of a single graphene layer so that the two layers are electronically decoupled. The twisted layer spectrum develops two Van-Hove singularities that flank the Dirac point at energies $\pm E \propto \sin \theta/2$. For $\theta > 10^\circ$ the Van-Hove singularities are separated by more than $1eV$ bringing them outside the energy range studied here [3, 5]. Experimentally, the two layers are stacked intentionally with a large twist angle [6]. The validity of the Dirac physics in the energy range of interest is directly confirmed by measuring the Landau level sequence in a magnetic field [6]. The purpose of using the double layer is to improve sample quality by isolating it from substrate induced potential functions as reported earlier [2], without at the same time destroying the Dirac physics. We have added a paragraph discussing this issue in the supplementary information for the benefit of a broader readership (page 4 of SM, left column).

In the second part of the question the reviewer points out that the Dirac physics may no longer be valid in the presence of the vacancy. This is indeed the case locally at the vacancy site. As was shown by ab-initio numerical simulations the vacancy site reconstructs and the on-site DOS develops a zero mode. As we have shown in Ref. 30 of the main text [6] the zero mode decays very rapidly away from the center of the vacancy and within a radius of $\sim 1nm$ the DOS characteristic of the massless Dirac electrons is fully recovered. However, once the charge exceeds the critical value, the resonances associated with the quasi-bound states extend over tens of nanometers in close agreement with theoretical predictions for the over critical quasi bound states of a massless Dirac particle in the presence of the Coulomb potential. Beyond this length scale the characteristic Dirac physics is fully recovered as confirmed by Landau level spectroscopy [6].

Figure 1: DOS maps [6] comparing the extent of the wave-function at a sub-critically charged vacancy (left panel) to that at a super-critically charged vacancy (right panel). The latter displays more than one order of magnitude increase in size.

As we explain in our text, the creation of a vacancy has an important and measurable consequence which is the breaking of the parity symmetry and the lifting of the degeneracy between the two

ladders of quasi-resonances in the overcritical regime (and the vanishing of the single quasi-resonance in the under-critical regime, see Fig. 2.a). We find that the existence and the β -behaviour of the E' peak(s) is a clear and qualitative experimental proof of this statement proven theoretically using massless Dirac physics.

2. **Secondly, the authors should include an actual atomic resolution STM scan to show how the atom positions actually look like in the sub critical and over critical situations, then the readers can easily judge whether the deformation is indeed a local confinement or some other wired form of classical confinement (which certainly will show similar change from a shallow trap with one state to deeper trap with multiple states).**

Regarding the first part of the question, STM topography is a convolution of the local DOS and height profile. It does not always show the position of atoms. In particular the characteristic STM signature of an isolated vacancy in graphene is the triangular interference pattern which arises due to the local crystal distortion and corresponding electronic state reconstruction. This is the feature that was used to identify single atom vacancies in this work. The STM topography did not reveal any distinguishable difference between the vacancies before and after the charge was deposited. We added the STM topography image to the SM (Fig. 4).

In the second part of the question, it is not clear what the referee means by 'classical confinement'. One of the intrinsic characteristics of massless Dirac fermions is their inability to be confined by electrostatic potentials. This is a consequence of their massless nature whose best known manifestation, Klein tunnelling, is also responsible for the extraordinarily high electron mobility in graphene. In the presence of a supercritical point charge, the Hamiltonian can admit a sequence of quasi-bound solutions as discussed in the main text and in earlier publications. But these states are qualitatively different and experimentally distinguishable from states that are caused by '**classical confinement (which certainly will show similar change from a shallow trap with one state to deeper trap with multiple states)**'. Such classical states as hypothesized by the referee are localized within the potential well and cannot extend far beyond its boundaries. In contrast, the overcritical quasi-bound states observed here extend far beyond the confines of the potential well as expected of resonances and in full agreement with the theoretical simulations [6]. Furthermore, had such states been present, they would have produced a spectral gap and a sequence of bound states with a characteristic energy $\approx \hbar v_F/D \approx 0.3eV$ (D the characteristic dimension of the well), which clearly is not observed in the experiment. A similar argument also rules out quantized levels due to Coulomb blockade physics $\approx e^2 v_F/\epsilon_0 D \approx 0.8eV$.

An important distinction between the overcritical and classical confinement states (such as in a quantum dot) is the spatial probability distribution of the wavefunction. In the former case there is a peak in the probability density at the center of the potential well for all states. This is in stark contrast to the quantum dot case where the first excited state (and all odd ones) has a node at the center of the well. Spatial STS maps are ideally suited to differentiate between the two possibilities by spatially mapping the DOS of the first excited state. Overcritical states show a bright spot at the center of the well while the classical confinement states are dark in the center as was demonstrated by [6]. This was clearly observed for the states reported here.

3. **Thirdly, I am not sure how the local charge is created. Since graphene is conducting, the local charge should disappear as soon as the tip voltage is removed, unless the charge is actually injected into the SiO2 substrate then it can be long lived and generate the readily tunable Coulomb potentials.**

Recent theoretical work using ab-initio calculations as well as experiments using Kelvin probe microscopy [7] have shown that the removal of a Carbon atom from the graphene honeycomb lattice produces a local charge imbalance of about +1 (units of electron charge) associated with the rearrangement of neighbouring sigma orbitals into a lower energy configuration. Although the ground state of the vacancy carries a local charge of +1, the removal of a Carbon atom by low energy He ion irradiation does not necessarily lead to the fully charged state. In fact we have shown by using Landau level spectroscopy that the initially prepared vacancies are trapped in a metastable state which carries very little charge [6]. It is only after applying voltage pulses at the vacancy site that the charge is gradually building up as the system approaches the ground state. We are carrying out experiments and DFT calculations for a future publication that will allow us to better understand and quantify the evolution of the vacancy charge as it approaches equilibrium.

Reviewer #3:

1. **The interpretation in terms of a QPT is novel and appears to qualitatively agree with the data presented. However, a careful reading leads to several questions that need to be addressed by the authors.**

We wish to stress that the agreement is not only qualitative but also quantitative. To make this point clearer, we added a new plot (Fig. 6) where we quantitatively compare the energy ratios to the universal factor $E_2/E_1 = e^{-\frac{\pi}{\sqrt{\beta^2-1/4}}}$. We find a very good agreement with this prediction. In addition we added to the main text a new paragraph corresponding to this plot (page 4, right column).

2. **Regarding the experimental part: As mentioned above, the experimental setup is a charged vacancy in a graphene sheet that realizes an attractive Coulomb potential for electrons. The procedure to reach the conditions of the phase transition requires further charging of the impurity to increase the effective strength of the potential. The increase is reached by applying multiple current pulses with the STM tip. A relevant question is: what occurs with screening? (incidentally: there is no reference accompanying the statement that ‘single atom vacancies in graphene can stably host local charge’).**

The question of screening in this problem was previously addressed theoretically by Shytov et al [8] as well as by other groups. These authors showed that the essential physics of this problem is not affected by screening. This is because at finite density the RPA screening length, which is comparable to the Fermi wavelength, is significantly smaller than the confinement radius of the quasi-bound state [8, 9]. Similarly, estimates for nonlinear screening [10] indicate that its effect is negligible at weak coupling, leaving enough room for quasi-bound states to form. We thank the reviewer for pointing out the omission of the reference. We have also added the reference [7] and a clarifying statement to the main text (page 3, right column).

3. **Regarding the theory interpretation there are two important points that need to be discussed:**

- (a) **the model is based on the analysis of a single (non-interacting) relativistic particle in terms of the Dirac equation in the presence of a Coulomb potential. The Coulomb potential that appears in Eq. 3 in the main text is proportional to $1/r$, where r is the radial coordinate in a polar coordinate system used to describe**

a 2d membrane. An electrical charge in graphene (either produced by a localized vacancy or an atom deposited on top or below) produces a 3d Coulomb potential (since it lives in a 3d world). I see two ways in which to consider this point: a) consider ‘standard’ spherical coordinates for the Coulomb potential and by proper integration obtain the effective 2d potential experienced by electrons in graphene, or b) use the generalized polar decomposition for Cartesian coordinates and separate radial from angular dependence, fixing the appropriate angular momenta eigenvalues by the constraint of 2d. Either of these procedures would render a different dependence on the polar radial/angular coordinates than the one assumed in the model.

Assuming the graphene sheet is a perfect plane defined (without loss of generality) by $z = 0$, and that the charge disturbance (vacancy) is at the origin of the 3d space, then the potential exerted by the charge is $V \sim \frac{1}{\sqrt{x^2+y^2+z^2}}$ and on the graphene sheet, $z = 0$, $V \sim \frac{1}{\sqrt{x^2+y^2+0^2}}$. Thus the spherical and polar coordinates coincide in this case, for which, indeed, the assumption of a two dimensional planar geometry is essential.

- (b) **Furthermore, although graphene is prepared on top of a substrate with a large twisted angle to achieve the weak coupling regime, there is screening produced by the substrate that is not even mentioned (I assume vacuum on top, thus the total screening is anisotropic).**

The contribution of the substrate (dielectric constant ≈ 4) to screening in the graphene plane was included by taking the average with the vacuum on top, to give 2. The substrate as well as the RPA contribution to screening from the carriers in graphene was discussed in detail theoretically [9] and also experimentally [1] in earlier publications.

- (c) **the role played by electron-electron interactions that are completely neglected. There seems to be a disconnect between the mathematical model that deals with a finite number of single particle bound states (akin to the spectra for trions or other similar charged impurities in semiconductors) and the experimental setup that creates a charged region of reduced dimensions (akin to a quantum dot or island). It would be expected that electron-electron interactions play an important role in experiment but there are not addressed by the theory model. Is screening the reason why they can be neglected? if not, why their effect is not included in the model?**

There are two aspects to the question. First, the relevance of electron-electron interactions in graphene is independent of the presence of the vacancy either charged or not. In the absence of the vacancy, the low energy quasi-particle excitations are well described by non-interacting 2d massless Dirac fermions, except for very close to the Dirac point where many-body effects lead to logarithmic corrections to the Fermi velocity. This regime however is inaccessible in most experimental situations [11].

In the second part the referee raises the possibility of attributing the observed resonances to quantum dot states that arise due to the charging energy or quantum confinement. However, one of the intrinsic characteristics of massless Dirac fermions is their inability to be confined by electrostatic potentials. This is a consequence of their massless nature whose best known manifestation, Klein tunnelling, is also responsible for the extraordinarily high electron mobility in graphene. In the presence of a supercritical point charge, the Hamiltonian can admit a sequence of quasi-bound solutions as discussed in the main text and in earlier publications. But these states are qualitatively different and experimentally distinguishable from states that are caused by quantum dot confinement. Such states as hypothesized by the referee are localized within the potential well and cannot extend far

beyond its boundaries. In contrast, the over critical quasi-bound states observed here extend far beyond the confines of the potential well as expected of resonances and in full agreement with the theoretical simulations [6]. Furthermore, had such states been present, they would have produced a spectral gap and a sequence of bound states with a characteristic energy $\approx \hbar v_F/D \approx 0.3eV$ (D the characteristic dimension), which clearly is not observed in the experiment. A similar argument also rules out quantized levels due to Coulomb blockade physics $\approx e^2 v_F/\varepsilon_0 D \approx 0.8eV$.

4. **It is not clear why the Sierpinski gasket is included in Fig. 1. I assume it is for pedagogical purposes but it would be more useful if instead more details in the experimental data are included. For example: Is the amount of charge transferred to the impurity controlled by the duration of the pulse? or is it the intensity of the current? How is the amount of charge transferred known?**

The purpose of Fig.1 is to present a visual abstract of the paper and to drag the attention of a broad readership. The Sierpinski gasket is included to substantiate the notion of discrete scale symmetry appearing in the essentially self-similar (fractal) spectrum in the over critical regime.

The amount of charge transferred through the vacancy cannot be tuned. Each pulse increases the previously injected charge by some positive amount that cannot be tuned beforehand. Fortunately, the charge increase is gradual enough so as to allow for the observation of the transition. As explained in the main text, the effective β resulting from the charge increase, is obtained by directly comparing the position of the experimental quasi bound states with the theoretical Dirac model.

The role of pulse duration and amplitude was studied in detail experimentally for hundreds of repetitions. Our findings are as follows:

- (a) We identified a window of voltage pulse amplitudes ($-2V < V < 2V$) within which the vacancy cannot be charged. Outside this window it is possible to charge the vacancy by applying voltage pulses. For the work presented here the pulse amplitude was confined to $2V$.
- (b) As long as the pulse duration exceeds 0.5s the charging process is observable.

5. **Being such a local defect, the vacancy must break the valley degeneracy locally, why it is assumed that parity remains a good symmetry for the interpretation of the data? It is natural to wonder if the degeneracy condition that needs to be broken for the peaks at E'_1 is not due to the breaking of valley degeneracy.**

We do not assume that parity is a good symmetry for the interpretation of the data. The experimental results reveal the existence of the E'_1 branch of peaks which precisely results from the lifted degeneracy associated with broken parity. This is a clear indication that our Dirac model must accommodate a parity breaking boundary condition. We wish to emphasise that the existence and relevance of the lifted degeneracy peaks E'_1 has not been addressed in the literature before. This is a novel experimental observation as well as theoretical interpretation. Parity transformation is equivalent to exchanging the two sub-lattices. Since the vacancy breaks this exchange symmetry of the sub-lattices, it is reasonable to expect that in the Dirac description parity must be broken as well. To make this point clearer we have added the sentence:

“In graphene, exchanging the triangular sub-lattices is equivalent to a parity transformation. Creating a vacancy breaks the symmetry between the two sub-lattices and is therefore at the origin of broken parity in the Dirac model” (page 5, left column).

6. **It is not clear what the STM images appearing in Fig. 1 represent. Do they represent the combined orbitals proposed in the theory model? If shown from the top would**

they look like the insets? The two figures have very different length scales: do they show the existence of a small ‘charged island’ after the transition is achieved? What is the reason for the Mexican hat-like structure?

Each STM images in Fig. 1 display the spatial dependence of dI/dV for a fixed value of β and at a fixed energy corresponding to a quasi-resonant peak. The left figure (d) is for $\beta < \beta_c$ and the right (e) is for $\beta > \beta_c$ with an energy corresponding to one of the E_1 quasi-resonances. The dI/dV maps display the spatial dependence of the probability density. Our purpose was to show in a very qualitative way the significant differences in shape and length scales between the spatial behaviour of the density below and above the transition. The ‘Mexican hat-like structure’ observed for $\beta > \beta_c$, results from the log-periodic oscillations (another fractal behaviour) expected for the wave function. These long range periodic oscillations must be contrasted to the spatially localised density visible in the under-critical regime.

Although such a characteristic behaviour is presented and explained in the SM, we chose not to detail it in the text for two reasons: (i) The data is insufficiently accurate to determine in a quantitative way the overcritical periodic oscillations. (ii) A qualitative comparison will unnecessarily clutter the text without really giving any new information

The insets represent analogue pictures deduced from the Efimov effect described using an effective Schrödinger equation with an attractive inverse square potential. This potential also accommodates a single bound state in the under-critical regime and a geometric series of bound states in the over critical regime. This geometric series is known to describe the bound states of a set of three atoms (Efimov effect). The fact that massless Dirac particles in graphene and the Efimov effect share the same spectrum is the universality we wish to point out in the paper and wish to hint about in Fig. 1. To make things clearer, we rephrased the lines on figures (e) and (d) in the caption of figure 1.

7. The comparison with the Efimov states with data from cold atom experiments is unclear. In the case of Efimov states the results are obtained for a 3d potential, while the fitting is done with a 2d potential. What is the nature of the connection?

The Efimov effect is a 3d non relativistic phenomenon described using an effective Schrödinger equation with an attractive inverse square potential. It is indeed very different from the (over-critical) quasi-resonant spectrum of massless Dirac excitations due to scattering by an electrically charged vacancy in 2d graphene. However, these two problems share in common their scale-free nature. The kinetic and potential energies scale both with the same inverse power of r , 2 for the Schrödinger problem and 1 for the Dirac problem. Remarkably, this property implies that both problems exhibit the same abrupt transition in which a **geometric series** spectrum appears in the over critical regime. It is this universality we wish to present in the paper, particularly in Fig. 5. Since the spectra are identical up to the ζ to β conversion and to an overall non universal constant that we scale away, we have been able to add on the $E_n(\beta)$ curves (see Figs. 5 and 4 in the SM) measured in graphene, the points obtained experimentally in two different cold atomic systems (caesium atoms and a heteronuclear Li-Cs mixture) and representing Efimov states. This emphasizes the claimed universality in a quantitative and visual way.

References

[1] A. Luican, G. Li, A. Reina, J. Kong, R. R. Nair, K. S. Novoselov, A. K. Geim, and E. Y. Andrei, Phys. Rev. Lett. **106**, 126802 (2011).
 [2] C.-P. Lu, M. Rodriguez-Vega, G. Li, A. Luican-Mayer, K. Watanabe, T. Taniguchi, E. Rossi, and E. Y. Andrei, Proceedings of the National Academy of Sciences **113**, 6623 (2016).

- [3] G. Li, A. Luican, J. L. Dos Santos, A. C. Neto, A. Reina, J. Kong, and E. Andrei, *Nature Physics* **6**, 109 (2010).
- [4] J. M. B. Lopes dos Santos, N. M. R. Peres, and A. H. Castro Neto, *Phys. Rev. Lett.* **99**, 256802 (2007).
- [5] I. Brihuega, P. Mallet, H. González-Herrero, G. Trambly de Laissardière, M. M. Ugeda, L. Magaud, J. M. Gómez-Rodríguez, F. Ynduráin, and J.-Y. Veillen, *Phys. Rev. Lett.* **109**, 196802 (2012).
- [6] J. Mao, Y. Jiang, D. Moldovan, G. Li, K. Watanabe, T. Taniguchi, M. R. Masir, F. M. Peeters, and E. Y. Andrei, *Nat Phys* **12**, 545 (2016).
- [7] Y. Liu, M. Weinert, and L. Li, *Nanotechnology* **26**, 035702 (2015).
- [8] A. V. Shytov, M. I. Katsnelson, and L. S. Levitov, *Phys. Rev. Lett.* **99**, 246802 (2007).
- [9] E. H. Hwang, S. Adam, and S. D. Sarma, *Phys. Rev. Lett.* **98**, 186806 (2007).
- [10] A. V. Shytov, M. I. Katsnelson, and L. S. Levitov, *Phys. Rev. Lett.* **99**, 236801 (2007).
- [11] A. Luican, G. Li, and E. Y. Andrei, *Phys. Rev. B* **83**, 041405 (2011).

REVIEWERS' COMMENTS:

Reviewer #1 (Remarks to the Author):

I have read both the revised manuscript and the accompanying letter and am completely satisfied with the authors' response to my comments. I recommend publication of this important paper without further delay.

Reviewer #2 (Remarks to the Author):

Though I still have reservations about the microscopic pictures how the charges are trapped, and how the deformed local environment could alter the proposed explanations, I do think the authors have addressed my questions satisfactorily. The results are good food for thought, and the readers can judge on their own. Thus I would recommend its publication in NC in the present form.

Reviewer #3 (Remarks to the Author):

Review report for NCOMMS1701883T

Ovdat et al.

I read the resubmitted version of the manuscript together with all the accompanying correspondence.

Overall, the authors have implemented several changes as suggested by all reviewers and have answered the points raised in my individual report. In regard to these specifically, I want to comment that, although the answer provided for my question about the fate of screening as the vacancy is charged was answered exactly in the opposite, it gave me the excuse to visit the original reference and confirm that the screening length is much larger than the length scale associated with these bound states. Thus, I find the answer provided to be satisfactory.